# Representing Partial Programs with Blended Abstract Semantics

**Maxwell Nye,**[*] **Yewen Pu, Matthew Bowers, Jacob Andreas, Josh Tenenbaum, Armando Solar-Lezama**
Massachusetts Institute of Technology

## Abstract

Synthesizing programs from examples requires searching over a vast, combinatorial space of possible programs. In this search process, a key challenge is representing the behavior of a partially written program before it can be executed, to judge if it is on the right track and predict where to search next. We introduce a general technique for representing partially written programs in a program synthesis engine. We take inspiration from the technique of abstract interpretation, in which an approximate execution model is used to determine if an unfinished program will eventually satisfy a goal specification. Here we *learn* an approximate execution model implemented as a modular neural network. By constructing compositional program representations that implicitly encode the interpretation semantics of the underlying programming language, we can represent partial programs using a flexible combination of concrete execution state and learned neural representations, using the learned approximate semantics when concrete semantics are not known (in unfinished parts of the program). We show that these hybrid neuro-symbolic representations enable execution-guided synthesizers to use more powerful language constructs, such as loops and higher-order functions, and can be used to synthesize programs more accurately for a given search budget than pure neural approaches in several domains.

## 1 Introduction

Inductive program synthesis – the problem of inferring programs from examples – offers the promise of building machine learning systems which are interpretable, generalize quickly, and allow us to solve structured tasks such as planning and interacting with computer systems. In recent years, neurally-guided program synthesis, which use deep learning to guide search over the space of possible programs, has emerged as a promising approach (Balog et al., 2016; Devlin et al., 2017). In this framework, partially-constructed programs are judged to determine if they are on the right track and to predict where to search next (see Figure 1). A key challenge in neural program synthesis is *representing* the behavior of partially written programs, in order to make these judgments. In this work, we present a novel method for representing the semantic content of partially written code, which can be used to guide search to solve program synthesis tasks.

Recently, approaches which represent partial programs via their *semantic* state have been shown to be particularly effective. In these **execution-guided neural synthesis** approaches (Chen et al., 2018; Ellis et al., 2019; Zohar & Wolf, 2018), partial programs are *executed* and represented with their return values. However, execution is not always possible for a partial program. In Figure 1, before the HOLE is filled with an integer value, we cannot meaningfully execute the partially-written loop in $s$. This is a common problem for languages containing higher-order functions and control flow, where execution of partially written code is often ill-defined. Thus, a key question is: How might we represent the semantics of unfinished code?

A classic method for representing program state, known as abstract interpretation, can be used to reason about the set of states that a partial program could reach, given the possible instantiations of the unfinished parts of the program. Using abstract interpretation, an approximate execution model can determine if an unfinished program will eventually satisfy a goal specification. However, this

---

[*]correspondence to `mnye@mit.edu`

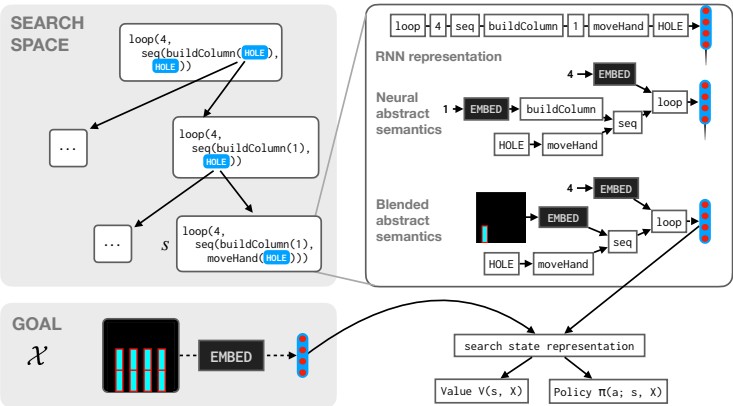

Figure 1: Schematic overview of the search procedure and representational scheme. We characterize program synthesis as a goal-conditioned search through the space of partial programs (left), and propose a novel representational scheme (blended abstract semantics) to facilitate this search process. Left: a particular trajectory through the space of partial programs, where the goal is to find a program satisfying the target image. Right: three encoding schemes for partial programs, which can each be used as the basis of a code-writing search policy and code-assessing value function.

technique is often low-precision; hand-designed abstract execution models greatly overapproximate the set of possible execution states, and contain no notion of what code is likely to be written.

We hypothesize that, by mimicking the compositional structure of abstract interpretation, learned components can be used to effectively represent ambiguous program state. In this work, we make two contributions: we introduce **neural abstract semantics**, in which a compositional, approximate execution model is used to represent partially written code. We further introduce **blended abstract semantics**, which aims to represent the state of unfinished programs as faithfully as possible by concretely executing program components whenever possible, and otherwise, approximating program state with a learned abstract execution model. This combination of learned execution and concrete execution allows robust representation of partial programs, which can be used for downstream synthesis tasks. We show that our model can effectively learn to represent partial program states for languages where previous execution-guided synthesis techniques are not applicable. In summary,

- We introduce blended neural semantics, a novel method for representing the semantic state of partially written programs inspired by abstract interpretation.
- We describe how to integrate our program representations into existing approaches for learning search policies and search heuristics.
- We validate our new approach with program synthesis experiments in three domains: tower construction, list processing, and string editing. We show that our approach outperforms neural synthesis baselines, solving at least 5% more programs in each domain.

## 2 BLENDED ABSTRACT SEMANTICS

Consider the problem of synthesizing arithmetic expressions from input–output pairs. Suppose we have the following context-free grammar for expressions:

$$\mathcal{G} = \texttt{E} \rightarrow \texttt{E} \ * \ \texttt{E} \ | \ \texttt{E} \ + \ \texttt{E} \ | \ \texttt{x} \ | \ \texttt{1} \ | \ \texttt{2} \ | \ \texttt{3} \ | \ \texttt{4}$$

and a specification $\mathcal{X}$ consisting of the input–output pairs $\{(x = 3, y = 7), (x = 5, y = 11)\}$. Suppose further that we have a candidate program $(\texttt{2} \ * \ \texttt{x}) \ + \ \texttt{1} \in \mathcal{G}$. To check that this program is consistent with the specification, we can evaluate it on the inputs $x$ in the specification according to the **concrete semantics** of the language. The goal of synthesis is to find an expression which satisfies the input-output examples under concrete semantics. To find such programs, we employ top-down search: starting with the top-level (incomplete) expression $\texttt{HOLE}$, we consider all possible expansions, $(\texttt{HOLE} \rightarrow \texttt{HOLE} \ + \ \texttt{HOLE}, \texttt{HOLE} \rightarrow \texttt{1}, \texttt{HOLE} \rightarrow \texttt{2}, \dots)$ and select the one we believe is most likely to succeed (Figure 1 left). The more effectively we can filter the set of incomplete candidate programs, the faster our synthesis algorithm.

Conventional **abstract interpretation** solves this problem by defining an alternative semantics for which even incomplete expressions can be evaluated. However, constructing appropriate abstrac-

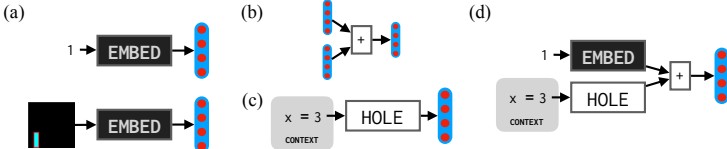

Figure 2: (a) Example applications of the EMBED function. (b) Neural abstract module for +. (c) Neural placeholder module encoding a HOLE with the context $\{x = 3\}$. (d) Neural abstract semantic encoding of the partial program `1 + HOLE` with the context $\{x = 3\}$.

tions is difficult and requires domain-specific engineering; an ideal procedure would automatically discover an effective space of abstract interpretations.

**Neural abstract semantics** $[\![\cdot]\!]^{nn}$   As a first step, we implement the abstract interpretation procedure with a neural network. (This is a natural choice: neural networks excel at representation learning, and the goal of abstract interpretation is to encode an informative representation of the set of values that could be returned by a partial program.) For the program `1 + HOLE`, we can encode the expression `1` to a learned representation (Figure 2a, top), likewise encode HOLE (Figure 2c), and finally employ a learned abstract implementation of the + operation (Figure 2b).

For concrete leaf nodes, such as constants or variables bound to constants, neural semantics are given using a **state embedding function** EMBED$(\cdot)$, which maps any concrete state in the programming language into a vector representation: EMBED : $(State \mid \mathbb{R}^d) \to \mathbb{R}^d$. If the input to EMBED is already vector-valued, EMBED performs the identity operation. **Neural placeholders** provide a method for computing a vector representation of unwritten code, denoted by the HOLE token. To compute the representation for HOLE, we define a neural embedding function $h$ which takes a context $\mathcal{C}$ and outputs a vector. For each built-in function $f$, the neural abstract semantics of a function $f$ are given by a separate **neural module** (a learned vector-valued function as in Andreas et al. (2016)) $[\![f]\!]^{nn}$ with the same arity as $f$. Therefore, computing the neural semantics means applying the neural function $[\![f]\!]^{nn}$ to its arguments, which returns a vector. Since the neural semantics mirrors the concrete semantics, its implementation does not require changes to the underlying language.

**Blended abstract semantics** $[\![\cdot]\!]^{blend}$   Notice that for an expression such as `(2 * x) + HOLE`, the concrete value of the sub-expression `(2 * x)` is known, since it contains no holes. The neural semantics above don't make use of this knowledge. To improve upon this, we extend neural semantics and introduce blended semantics, which alternates between neural and concrete interpretation as appropriate for a given expression:

- If the expression is a constant or a variable, use the concrete semantics.

- If the expression is a HOLE, use the neural semantics.

- If the expression is a function call, recursively evaluate the expressions that are the arguments to the function. If all arguments evaluate to concrete values, execute the function concretely. If any argument evaluates to a vector representation, transform all concrete values to vectors using EMBED and apply the neural semantics of the function.

Because blended abstract semantics replaces concrete sub-components with their concrete values, we expect blended semantics to result in more robust representations, especially for long or complex programs where large portions can be concretely executed. See the appendix for more details.

**Synthesis**   To perform synthesis, we experiment with methods to guide search introduced in Ellis et al. (2019). In this work, the search over partial programs is formulated as an MDP, in which each state is a pair $(s, \mathcal{X})$ consisting of a partial-program and a specification, and actions $a \in \mathcal{G}$ are expansions of HOLEs under rules under the grammar. We assume a reward of 1 for programs which satisfy $\mathcal{X}$. In this framework, we **learn to search** by training a policy $\pi(a|s, \mathcal{X})$ that proposes expansions to $s$, and optionally a value function $V(s, \mathcal{X})$ that predicts the probability that $\mathcal{X}$ is solvable via any expansion of $s$. Details can be found in the appendix. At test time, we explore a variety of code-writing search algorithms. Using only a policy, we can employ sample-based search and best-first search (where the log probability of generating $s$ under $\pi$ is used as the scoring function). With the addition of a learned value function, we can perform A*-based search with $-\log V(s, \mathcal{X})$ as a heuristic (see Ellis et al. (2019) for details).

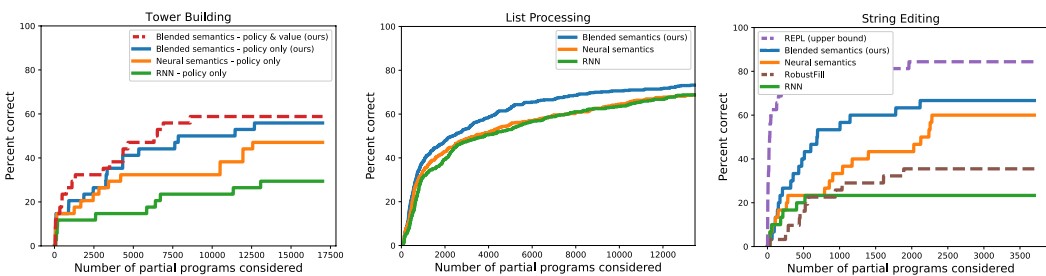

Figure 3: Synthesis results. Left: Tower building. Middle: List processing. Right: String editing. Blended abstract semantics outperforms baselines in synthesis tasks in each domain.

## 3 EXPERIMENTS

We evaluate our model in two domains containing language constructs not handled by concrete execution-guided synthesis approaches: a tower-building domain with looping constructs, and a list-processing domain with higher order functions. We additionally test on a string-editing domain for which execution-guided synthesis is possible, but requires extensive DSL modification; there, we examine how our approach fares without these modifications (see appendix).

**Looping constructs: Tower construction**    We begin by investigating how our model performs in generative programming domains with higher-level control flow such as loops. Looping programs are an essential part of sophisticated programming languages, and aren't naturally handled by previous execution-guided synthesis approaches. Our experiments in the tower-building domain employ a DSL similar to the language depicted in the introduction to construct towers in a 2D world. As above, the goal is to construct a program which successfully renders to a target image (examples in Figure 4). Language details can be found in the appendix. We compared against two baselines (see Figure 1): (1) Neural abstract semantics (defined in Section 2), which does not apply concrete execution to concrete subtrees, and (2) RNN encoding, which encodes partial programs using a GRU: $\pi(a \mid \texttt{GRU\_enc}(s), \mathcal{X})$. To evaluate our model, we constructed a test set of tower-building problems involving combinations of tower-building motifs seen during training. We evaluate our models by performing best-first search from the learned policy. We also test using a value function, where we are doing A* search with the policy as the prior cost and the value function as the heuristic future cost estimate. Figure 3 left shows our overall synthesis results in the tower-building domain, measuring the percentage of test problems solved as a function of the number of search nodes (partial programs) considered. The sequence encoding performs poorly and is unable to solve a majority of test problems. The neural abstract semantics model achieves better performance, solving about half of the test problems within the allotted search budget. Blended execution outperforms both baselines. We additionally observe that adding a value function as a search heuristic further increases performance of our blended model, which is consistent with the findings in Ellis et al. (2019).

**Higher-order functions: functional list processing**    In our second experimental domain, we seek to answer two questions: How well does our model perform on input-output synthesis? How effectively can it synthesize programs containing higher-order functions? Although previous work (Zohar & Wolf, 2018) has successfully applied execution-guided approaches to list processing (using the DeepCoder language), the use of higher-order functions was severely limited: only a small, predefined set of "lambdas," (such as $(\ast 2)$, $\texttt{is\_even}$, $(>0)$) were used as arguments for higher-order functions. For example, synthesizing a program which "filters all elements divisible by 3 from a list" is not possible with this DSL. However, in real programming languages, higher-order functions must be able to accept a combinatorially large set of possible lambda functions as input. This presents a challenge for execution-guided synthesis approaches such as Zohar & Wolf (2018), for which the assumption of a small set of lambda functions is key. To this end, we modified the Deepcoder DSL to allow a richer set of possible programs. We replaced the predefined set of lambda functions with a grammar allowing for the combinatorial combination of grammar elements (examples in Figure 5 left in the appendix). The modified grammar is given in the appendix. Figure 3 middle shows the results of synthesis using best-first search from a policy on test problems sampled the same distribution as the training problems. Our blended model finds the highest overall number of correct programs, achieving 5-10% higher accuracy given the same search budget compared to the neural semantics and RNN encoding schemes. The blended model also yielded superior results

on numerous variations of these tasks (increasing number of higher order functions, varying integer ranges, varying search method, etc). See the appendix for details.

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

## A  RELATED WORK

Synthesizing programs from examples is a classic AI problem which has seen advances from the Programming Languages community (Solar-Lezama, 2008). Recently, much progress has been made using neural methods to aid search. Such techniques include enumerative approaches (Balog et al., 2016), translation-based techniques (Devlin et al., 2017), hybrid approaches using sketches such as Murali et al. (2017), Nye et al. (2019), and Dong & Lapata (2018), among others. The recent work of Odena & Sutton (2019) also studies program representation, proposing property signatures. Work has also been done using graph neural networks to encode the syntax of programs (Allamanis et al., 2018; Brockschmidt et al., 2018) for bug fixing, variable naming, and synthesis.

Recent work has introduced the notion of "execution-guided neural program synthesis" (Ellis et al., 2019; Chen et al., 2018; Zohar & Wolf, 2018). In this framework, the neural representations used for search are conditioned on the executed program state instead of the program syntax. These techniques have been shown to solve difficult search problems outside the scope of enumerate or syntax-based neural synthesis alone. However, such execution-guided approaches have several limitations. We aim to generalize execution guided synthesis, so that it can be applicable to a wider range of domains, search techniques, and programming language constructs.

Our work is directly inspired by two techniques: abstract interpretation-based synthesis (Wang et al., 2017; Hu et al., 2019) and neural module networks (Andreas et al., 2016; Johnson et al., 2017). We employ module networks to implement blended neural execution, which aims to provide a learned execution scheme directly inspired by abstract interpretation. This approach is also related to other tree-structured encoders (Socher et al., 2011; Dyer et al., 2016).

| List Processing | | String Editing | |
|---|---|---|---|
| Examples | Program | Examples | Program |
| `[-4,9,4,6] → [18,12]` `[15,3,3,-14] → [30,6,6]` | `map (λx.x*2) (filter` `(λx.(x>0 & x%3==0)) input)` | `+106 769-858-438 → (769)` `+63 099-824-351 → (099)` | `Const('(') | GetToken(Number, 1)` `| Const(')')` |
| `[1,2,3,4] → [5,5,5,5]` `[1,2,4,-1] → [0,6,6,0]` | `zipwith (λx,y.x+y)` `(input) (reverse input)` | `Mariya Sergienko → Dr. Mariya` `Andrew Cencici → Dr. Andrew` | `Const('D') | Const('r')` `| Const('.') | Const(' ')` `| GetToken(Word, 0)` |

Figure 5: Example programs from the list processing (left) and string editing (right) domains.

## B  PROGRAM SYNTHESIS WITH BLENDED ABSTRACT SEMANTICS

Let $\mathcal{X} = \{(x_i, y_i)\}$, where $(x_i, y_i)$ are input-output pairs. Let $[\![s]\!]^{blend}_{x_i}$ denote the blended abstract semantic representation of $s$ with input $x_i$. The representation of a state $\mathrm{rep}(s, \mathcal{X})$ is:

$$\mathrm{rep}(s, \mathcal{X}) = \frac{1}{n} \sum_{x_i, y_i \in \mathcal{X}} \mathrm{ReLU}(W([\![s]\!]^{blend}_{x_i}; \mathrm{EMBED}(y_i)]))$$

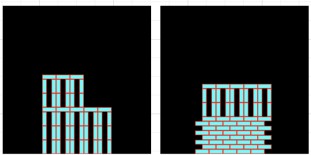

Figure 4: Constructions in the tower-building domain.

Here, $W$ is a learnable weight matrix, and the representation is averaged across all input-output pairs of $\mathcal{X}$. Given this state representation, the **policy** and **value function** are defined as follows:

$$\pi(a \mid s, \mathcal{X}) = \mathrm{softmax}(\mathrm{MLP\_a}(\mathrm{rep}(s, \mathcal{X})))$$
$$V(s, \mathcal{X}) = \sigma(\mathrm{MLP\_V}(\mathrm{rep}(s, \mathcal{X})))$$

Here, `MLP` is a multi-layer perceptron. Note the value function outputs a value between $0$ and $1$; this allows for a probabilistic interpretation.

**End-to-end training**  We train our policy $\pi$ using imitation learning. Starting from the empty partial program $s_0 = \mathrm{HOLE}$, we generate a sequence of partial programs $s_1, s_2, \cdots$ by sampling a sequence of expansions $a_0, a_1, \cdots$ from the grammar $\mathcal{G}$. Let $p = s_T$ be the completed program. We obtain specifications $\mathcal{X} = \{(x_i, y_i)\}$ by sampling a set of inputs $x_1 \cdots x_n$ and obtaining outputs using concrete semantics $y_i = [\![p]\!]^c_{x_i}$. Thus, from a sequence of expansions $a_0, a_1, \cdots$, we can

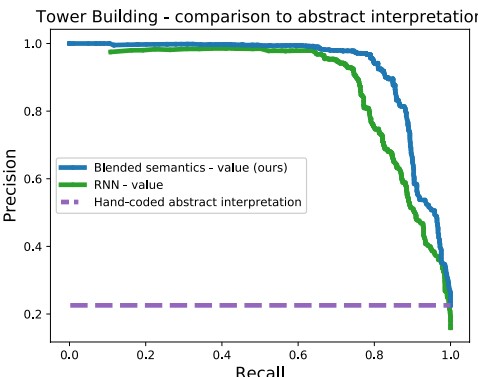

Figure 6: Comparing the value function to a hand-coded abstract interpretation. Blended abstract semantics outperforms baselines in synthesis tasks, and obtains higher classification precision than hand-coded abstract interpretation.

collect a set of triplets $\{(\mathcal{X}, s_i, a_i)\}$ as training data. This process is repeated to generate the training set $\mathcal{D}$. We can then perform supervised training, maximizing the log likelihood of the following:

$$\mathcal{L}(\pi) = \mathbb{E}_{(\mathcal{X}, s, a) \sim \mathcal{D}} \left[ \log \pi(a \mid s, \mathcal{X}) \right]$$

We train the value function by sampling rollouts of partial programs $s_0 \cdots s_T$ from a fully-trained $\pi$, minimizing the error in a Monte-Carlo estimate of the expected reward $R$ (i.e., the probability of success under the policy).

$$\mathcal{L}^{\text{RL}}(V) = \mathbb{E}_{(R, s_0 \dots s_T) \sim \pi(\cdot \mid s_0, \mathcal{X})} \left[ \sum_{t \leq T} \text{err}(V(s_t, \mathcal{X}), R) \right]$$

For our error function, we use a logistic loss rather than the more common MSE.

## C    COMPARISON TO ABSTRACT INTERPRETATION

How does the learned value function compare to hand-coded abstract interpretation? Classic abstract interpretation is conservative; it can be thought of as a classifier with perfect recall, but poor precision, only rejecting the partial programs it knows for sure to be unsuitable. Can our value function also detect these clearly bad partial programs, but ascribe low value to less obviously bad candidates? To test this, we performed an experiment in the tower-building domain. We conditioned the model on tasks from our test corpus, and sampled 15 search trajectories from our blended semantics policy for each task. For each partial program encountered during search, we compute the model's value judgment, and recorded whether each rollout was successful. Treating rollout success as a noisy label of partial program quality, and using the value function as a classifier, we plot precision vs recall of the value judgements as we vary the classification threshold. Figure 6 shows our results for this experiment. As the classification threshold is varied, our learned value maintains comparable recall compared to the hand-coded abstraction, while achieving better precision. For high classification thresholds, our model achieves performance comparable to the hand-coded abstract interpretation, and additional precision is gained by lowering the classification threshold. The RNN value performs worse on this test, achieving lower precision and recall.

## D    IO PROGRAMMING: STRING EDITING

In an additional experiment, we examine how our model performs on domains for which execution-guided synthesis is possible, but requires extensive changes to the underlying DSL.

For example, in the RobustFill DSL, a function `getSubStr(i,j)` slices a string from index $i$ to index $j$. This function is not executable until both $i$ and $j$ are known. In order to perform execution-guided synthesis, Ellis et al. (2019) needed to replace `getSubString` with two separate functions:

`getSubStrStart_i` and `getSubStrEnd_j`, where each half can be executed in the REPL. This process must be performed manually for every language construct which takes multiple arguments.

Here we seek to answer the question: can our model be used to successfully synthesize programs using the language as-is? To this end, we implement the code-writing policy using the DSL presented in Devlin et al. (2017) without modification (example programs in Figure 5 left). We do not expect that our approach would outperform the REPL system in Ellis et al. (2019), but we hope that it could achieve much of the gains. We additionally compare against another relevant baseline: RobustFill (Devlin et al., 2017). In contrast to the original paper, we train the RobustFill model using the same "unmodified" version of the DSL as our model, whose syntax has not been modified to aid with prediction. At test time, we used a sample-based search procedure, because the branching factor is prohibitively large for breadth-first search procedures explored above.

While the blended encoding does not achieve the accuracy of the execution-guided REPL system, it outperforms the other baselines, including the RobustFill model, neural abstract semantics and the RNN baseline.

## E  SEMANTICS

Here we fully define the semantics for concrete, neural, and blended semantics, covering details that were omitted in the main paper.

**Concrete semantics**   In this work, we consider domains where the underlying programming language is *functional*. Let $\lambda x.E$ be a lambda expression of one argument, and let $x = v$ be an assignment of the variable $x$ to the value $v$. Lambda application is defined as follows:

$$(\lambda x.E)(v) = [\![E]\!]_{\{x=v\}}$$

That is, we evaluate the function body $E$, replacing all instances of the function variable $x$ with the value $v$. For example, $[\![(\lambda x.x + x)(5)]\!]_{\{\}} = [\![x + x]\!]_{\{x=5\}} = 5 \hat{+} 5 = 10$. Where $\hat{+}$ denotes the execution semantics of the built-in function $+$. The concrete semantics $[\![\cdot]\!]$ is defined:

$$[\![E]\!]_{\mathcal{C}} = \begin{cases} [\![k]\!]_{\mathcal{C}} = k & \text{a constant } k \\ [\![x]\!]_{\mathcal{C}\models x=v} = v & x = v \\ [\![f(E_1\cdots)]\!]_{\mathcal{C}} = \hat{f}([\![E_1]\!]_{\mathcal{C}}\cdots) & \text{executing built-in } f \\ [\![(\lambda x_1\cdots x_n.E)(E_1,\cdots,E_n)]\!]_{\mathcal{C}} = [\![E]\!]_{\mathcal{C}\cup\{x_1=[\![E_1]\!]_{\mathcal{C}},\cdots,x_n=[\![E_n]\!]_{\mathcal{C}}\}} & \text{lambda application} \end{cases}$$

**Neural semantics**   For built-in functions $f$, we use $f^{nn}$, a neural module function of $k$ vector inputs, where $k$ is the arity of $f$.

In our domains, lambda expressions are only used as arguments to higher-order functions. Therefore, since we will never apply a lambda expression directly under neural semantics, we only require vector representation of lambdas. However, we still require a mechanism to represent arbitrary lambda expressions built combinatorially from primitive functions. This representation is constructed in a modular fashion by encoding the body $E$ of the lambda expression.

$$[\![E]\!]_{\mathcal{C}}^{nn} = \begin{cases} [\![k]\!]_{\mathcal{C}}^{nn} = \text{EMBED}(k) & \text{a constant } k \text{ is embedded} \\ [\![x]\!]_{\mathcal{C}\models x=v}^{nn} = \text{EMBED}(v) & \text{embed the value } v \text{ of the variable } x \\ [\![\text{HOLE}]\!]_{\mathcal{C}}^{nn} = h(\mathcal{C}) & \text{a neural placeholder for a hole based on context} \\ [\![f(E_1\cdots)]\!]_{\mathcal{C}}^{nn} = f^{nn}([\![E_1]\!]_{\mathcal{C}}^{nn}\cdots) & \text{using neural module} \\ [\![\lambda x.E]\!]_{\mathcal{C}}^{nn} = [\![E]\!]_{\mathcal{C}\cup\{x=\text{null}\}}^{nn} & \text{encode the body of a lambda} \end{cases}$$

Note that in representing a lambda expression, we used the context with the assignment $x = \text{null}$. This is used to account for the fact that, in the time of the lambda function's definition (as an argument to a higher-order function), its argument is still unknown under neural execution.

The definition for blended semantics proceeds in an analogous fashion, with concrete subtrees executed concretely.

## F   EXPERIMENTAL DETAILS

All models are trained with the AMSGrad (Reddi et al., 2018) variant of the Adam optimizer with a learning rate of 0.001. All RNNs are 1-layer and bidirectional GRUs, where the final hidden state is used as the output representation. Unless otherwise stated, holes are encoded by applying the EMBED function to the context, and then applying a type-specific neural module to the resulting vector.

### F.1   TOWERS

We employ the tower-building domain and DSL introduced in Ellis et al. (2020), which consists of the basic commands: `PlaceHorizontalBlock`, `PlaceVerticalBlock`, `MoveHand(n)`, `ReverseHand()`, `Embed`, `Loop` and integers n from 1 to 8. (The higher-order `Embed` function takes an expression as input, executes it, and then returns the hand to its initial location.) In order to test the compatibility of our approach with library-learning techniques, we additionally use library functions learned by the DreamCoder system by combining the above functions. Following Ellis et al. (2020), the grammar is implemented in continuation-passing style. Our training data consisted of tower programs randomly sampled from a PCFG generative model (Ellis et al., 2020).

We trained policy networks on 480000 programs. We trained value functions on 240000 rollouts from the policy. We perform search for up to 300 seconds per problem.

All neural modules consist of a single linear layer (input dimension $512 * n_{args}$ and output dimension 512) followed by ReLU activation. Tower images are embedded with a simple CNN-ReLU-MaxPool architecture, as in Ellis et al. (2020).

**Hand-coded abstract interpretation**   We implemented an abstract domain which tracked, a) the range of possible locations of the "hand" and b) for each horizontal location, the minimum height which must be achieved by the partially constructed tower. This representation allows us to eliminate invalid partial programs because once a block is dropped, it cannot be removed through any subsequent commands.

### F.2   LIST PROCESSING

Data for this domain was generated by modifying the DeepCoder dataset (Balog et al., 2016). Specifically, DeepCoder training programs of size 2 (containing 2-3 higher order functions, such as `map f (filter g input)` or `zipwith f (map g input) (map h input)` ) were modified by changing the lambdas in the program (`f`, `g`, and `h` in the above examples) from a small set of constant lambdas such as `(*2)` to depth-3 lambdas sampled from our modified grammar (see below). For example: `(λx.max(x+2,x/2))`. For each program, 5 example input lists were sampled, each with length 10 and values in the range [-64, 64]. The program was then executed to yield the corresponding outputs. Programs with output or intermediate values outside of the range [-64, 64] were discarded. Programs producing the identity function or constant functions were also discarded. We trained and tested only on functions of type `[int]` → `[int]`. At test time when running search, we similarly reject programs with intermediate values outside of the desired integer range.

All policy networks were trained on 500000 programs. We perform search for 180 seconds per problem.

All neural modules consist of a single linear layer (input dimension $64 * n_{args}$ and output dimension 64) followed by ReLU activation. Integers are encoded digit-wise via a GRU. Lists are encoded via a GRU encoding over the representations of the integers they contain.

Unbound variables within a lambda function are embedded via a learned representation parameterized by the variable name (one vector representing x and one representing y). When encoding holes within lambda functions, we ignore context, and instead embed holes only as a function of the hole type.

**Modified lambda grammar**   Below is the grammar used for lambda functions:
`L → (λx,y.S) | (λx.S)`

```
S → I | B
I → I+I | I*I | I/I | min(I,I) | max(I,I) | A
B → I > I | or(B,B) | and(B,B) | I%I==0
A → x | y | N
N → -2 | -1| 0 | 1 | 2
```

**Variations on training and testing conditions**    Many variations on the training and testing conditions achieve similar results to those shown in the main paper (i.e., blended semantics consistently achieves the highest performance). Several of these variations are shown in Figure 7.

### F.3   STRING EDITING

For the string editing tasks, we use the DSL from Devlin et al. (2017). We train on randomly sampled programs, sampling I/O pairs and propagating constraints from programs to inputs to ensure that input strings are relevant for the target program (see Devlin et al. (2017)). We condition on 4 I/O examples for each program. We used string editing problems from the SyGuS (Alur et al., 2016) program synthesis competition as our test corpus.

We trained all models on 2 million training programs. At test time, we sample programs from the model for a maximum timeout of 30 seconds.

Input and output strings are encoded by embedding each character via a 20-dimensional character embedding and concatenating the resulting vectors to form a representation for each string. Representations of "expressions" $e$ (as defined in the RobustFill DSL) are concatenated together using an "append" module. Following Ellis et al. (2019), neural modules consist of a single dense block with 5 layers and a growth rate of 128 (input dimension $256 * n_{args}$ and output dimension 256).

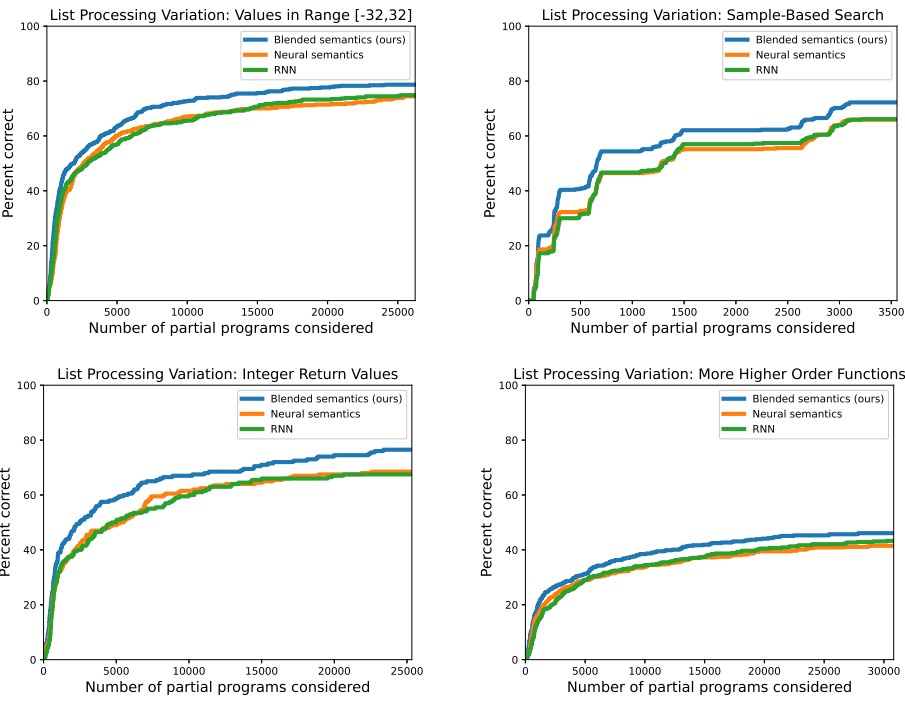

Figure 7: Variations on the list processing task. Top Left: using integer values in the range [-32, 32] instead of [-64, 64]. Top Right: Using sample-based search instead of best-first search. Bottom Left: extending the training and testing data to allow for [int] → int functions. Bottom Right: the original model tested on deeper DeepCoder programs with 3-6 higher-order functions. Notably, in all test conditions, the blended semantics consistently outperforms the RNN and neural semantics.