# OpenReview forum: "Representing Partial Programs with Blended Abstract Semantics"
_NeurIPS.cc/2020/Workshop/CAP — NeurIPS 2020 CAP Workshop_

### Official Review · AnonReviewer1 · 2020-10-31
**Great program induction paper, but the analogy to abstract interpretation needs some follow-up work**

**Rating:** 7
**Confidence:** 5

**Review:**

The paper proposes an approach to neural program synthesis from I/O examples that builds a neural approximation to the semantics of the current partial program under consideration. This approximation, coined blended abstract semantics (BAS), consists of an embedding function for every constant/variable/hole and a learned neural module network for every operator. The fully realized parts of the program are evaluated concretely before embedding. This program representation acts as a state conditioning for a trained policy that builds the desired program step by step using best-first or A* search. The approach shows significant improvement over multiple program induction baselines.

I am intrigued by the parallel to abstract interpretation and find this approach to program representation clearly effective for guiding a search in program induction tasks. As such, the paper is definitely interesting and worthy of acceptance and discussion. However, I feel like the analogy is stretched a bit too far here. Consider:

1)
As an "abstract interpretation" analogue, BAS can only move "up" the (imaginary) lattice. During bottom-up computation, the presence of any hole renders the whole expression containing it abstract. New information (e.g. multiple examples, the target spec, or PL rules) cannot collapse the representation back into the concrete space, it only (hopefully but uninterpretably) impacts the representation in the embedding space analogously.

Simple example: 0*<HOLE> will always be represented abstractly (rather than concretely as 0) using the naïve description in the paper. This situation is trivial to detect in the implementation but more nuanced versions of it are easy to construct.

As a more substantial example, any situation where multiple examples are needed to resolve ambiguity doesn't manifest itself in the program representation but only in the policy network. This is because the BAS representation is only conditioned on a single input at a time. For example:

    Candidate program: filter (λx → x % <HOLE> == 0) L
    Spec: L=[6, 1, 12] → [6, 12]; L=[15, 7] → [15]

Either example is inconcrete on its own, but both of them unambiguously resolve <HOLE> to 3. Such resolution might be learned by the policy network (as evident from the method's high performance), but only if the BAS representation of each <program, example> pair is informative enough. Which brings me to my second point:

2)
Since NMNs and the policy network are trained jointly, their embedding manifolds co-adapt. The embedding space shaped by the learned NMNs cannot be interpreted by any means other than the behavior of π. Thus, it's not exactly fair to call BAS an "abstract interpretation" of the partial program. Rather it's BAS plus its usage by π that can be characterized as such. Put another way, the classic question of abstract interpretation "Is this state theoretically reachable by this partial program P?" cannot be answered by looking at BAS(P) alone, we must observe the search behavior to do it. Which is what the authors essentially do in Appendix C.

I suppose the learned BAS and the policy network could be disentangled if one added a secondary objective that somehow regularized the embedding space. For example, it could be a simple abstract-interpretation classifier directly on top of the produced program representations. (An interesting experiment for the next version of the paper.)

3)
Finally, BAS, in contrast to only NAS, loses information about the syntax of the program when it performs "constant folding" (i.e. concrete evaluation). This does not matter for the setting of guiding program induction where we consider any program satisfying the example spec correct. However, for other program synthesis settings (e.g. likely completion in a context, semantic parsing from natural language, quantitative synthesis with a cost function) representing the program's syntax jointly with its behavior matters. In NAS that can be achieved implicitly, as every learned NMN can in theory propagate a representation of the whole AST. In BAS, however, information is lost during concrete evaluation.

One simple fix could be to define:

    BAS( f(concrete_expr, abstract_expr) ) = [[f]](H, BAS(abstract_expr))
    where
      [[f]] is the learned NMN for f,
      H = Concat( Embed(ConcreteEval(concrete_expr)); NAS(concrete_expr) )

rather than current `H = Embed(ConcreteEval(concrete_expr))`.

In conclusion, please don't interpret my comments as negative feedback. The idea of neural abstract semantics is neat and using it to guide program induction search is a great and natural extension of execution-guided program synthesis literature. I wholeheartedly champion the paper. Rather, I'm suggesting ways to improve presentation and strengthen the paper so that its "abstract interpretation" parallel does not overclaim and distract from the core contributions of the work.

---

### Decision · Program_Chairs · 2020-11-02

**Decision:**

Accept

**Comment:**

The reviewer is clearly in favor of accept and I agree with their recommendation.